# Lactobacilli Supplemented with Larch Arabinogalactan and Colostrum Stimulates an Immune Response towards Peripheral NK Activation and Gut Tolerance

**DOI:** 10.3390/nu12061706

**Published:** 2020-06-07

**Authors:** Tsvetelina Velikova, Kalina Tumangelova-Yuzeir, Ralitsa Georgieva, Ekaterina Ivanova-Todorova, Elena Karaivanova, Ventsislav Nakov, Radislav Nakov, Dobroslav Kyurkchiev

**Affiliations:** 1Clinical Immunology, University Hospital “Lozenetz,” Kozyak 1 Str., 1407 Sofia, Medical Faculty, Sofia University St. Kliment Ohridski, 1504 Sofia, Bulgaria; 2Laboratory of Clinical Immunology, University Hospital “St. Ivan Rilski”, Medical University of Sofia, 15 “Acad. Iv. Evst. Geshov” Blvd, 1431 Sofia, Bulgaria; kullhem000@gmail.com (K.T.-Y.); katty_iv@yahoo.com (E.I.-T.); dsk666@gmail.com (D.K.); 3Lactina Ltd., 101 Sofia str., 1720 Bankya, Bulgaria; ralitsa.georgieva@lactina-ltd.com (R.G.); elena.karaivanova@lactina-ltd.com (E.K.); 4Clinic of Gastroenterology, University Hospital “Tsaritsa Ioanna–ISUL,” 8 Byalo more Str., 1527 Sofia, Bulgaria; dr.nakov1962@abv.bg (V.N.); radislav.nakov@gmail.com (R.N.)

**Keywords:** *Lactobacillus* spp., larch arabinogalactan, colostrum, mucosal cytokines, NK cells, probiotic, prebiotic

## Abstract

Probiotics possibly affect local and systemic immune reactions and maintain the intestinal immune homeostasis in healthy individuals and patients with diseases such as irritable bowel syndrome (IBS). In this single-center, blinded trial, we enrolled 40 individuals (20 patients with IBS and 20 healthy individuals) whose blood and fecal samples were collected before and after a 21-day administration of a product comprising *Lactobacillus* spp., larch arabinogalactan, and colostrum. The percentage of HLA-DR+ natural killer (NK) cells was higher in healthy individuals (*p* = 0.03) than in patients with IBS after product supplementation. In the fecal samples of patients with IBS, we observed a decline in IL-6, IFN-γ, TNF-α, and secretory IgA levels and, simultaneously, an increase in IL-10 and IL-17A levels after supplementation, although non-significant, whereas, in healthy individuals, we observed a significant decline in IL-6 and IFN-γ levels after supplementation (*p* < 0.001). Nevertheless, we observed a clinical improvement of symptoms in 65–75% of patients with IBS and the complete resolution of the initial symptoms in five of the 20 patients. We also observed a possible prophylactic effect by the inducing system antiviral impact accompanied by a trend for local immune tolerance in the gut in healthy individuals, where it is the desirable state.

## 1. Introduction

The gut microbiota is crucial for the development of a proper immune system in the intestinal lumen. Thus, the microflora plays an essential role in the maintenance of intestinal immune homeostasis. However, the growing prevalence of the Western diet as a part of the modern lifestyle may favor a transition in the microbial composition, leading to dysbiosis [1]. Dysbiosis is associated with not only gastrointestinal discomfort but also healthcare problems such as inflammatory bowel disease, diabetes, and obesity [2]. However, many other factors may influence the gut microbiome other than diets, such as environmental and genetic factors, smoking, alcohol use, lifestyle, stress, drug use (especially antibiotics), the minerals contained in drinking water, etc. [3]. All of these may disrupt the normal microbiome.

In line with this, several animal and human studies have focused on manipulating the gastrointestinal microbiota with diet and probiotics [4]. Since the introduction of the term “probiotics” by Metchnikoff around 1900, several definitions have been used. Currently, the widely accepted definition of probiotics is “live microorganisms that may comprise different bacterial strains, which, when consumed in suitable amounts, deliberate a health effect on the host” [5]. Probiotics affect not only local and systemic immune reactions but also exert various effects on the intestinal mucosa, including the barrier function and synthesis of antimicrobial peptides [6]. Moreover, probiotics become a part of the healthy microflora in the recipient’s gut, representing an anatomical–functional unit that modulates the cell-mediated and humoral immune responses and the local production of cytokines through the mucosa-associated lymphoid tissue system.

In contrast, prebiotics, firstly described as nondigestible food ingredients that selectively stimulate the growth and/or activity of certain bacteria in the colon, therefore improving digestive and host health, were later precisely defined as selectively fermented ingredients that allow specific changes, including composition and/or activity, in the gut microbiome, providing benefits for the host [7]. However, dietary fibers may exert host benefits beyond gastrointestinal health, including improvement of the cardiovascular system, diabetes, appetite control, and body weight. Nevertheless, due to their variable effects in the body, it is recommended to consume fibers from a variety of sources [7].

Arabinogalactans are a source of dietary fibers that may stimulate the growth of beneficial intestinal microflora and, thus, support digestive health. While arabinogalactans are found in various plants, they are more abundant in the larch tree (Larix species) [8]. A combination of arabinogalactans and probiotics may exert synergistic effects and increase the efficacy of the microbial strains included. Lately, larch arabinogalactans have gathered considerable attention as clinically useful nutraceutical agents because of their potential therapeutic benefits as immune-enhancing agents. In addition, these can exert their effect indirectly via microbiota-dependent mechanisms (relying on the action of short-chain fatty acids (SCFAs)) and directly on gut-associated immunity in the small intestine after passage through the gut-associated lymphoid tissue [9].

Irritable bowel syndrome (IBS) is the most common among functional gastrointestinal disorders that comprise conditions primarily explained as “disorders of the gut–brain interaction” [10]. The existing concepts of the pathogenesis of IBS focus on potential alterations in the gut motility, small-bowel bacterial overgrowth, low-grade microscopic inflammation throughout the small bowel and colon, immune activation, visceral hypersensitivity, changes in the brain–gut axis [11], and disruptions in the gastrointestinal microbiota [12,13]. In addition, gut microbiota plays a role in the pathogenesis of IBS because IBS is more frequently observed after an intestinal infection or antibiotics treatment. However, the beneficial effects of modulating the gut microbiota using probiotics in patients with IBS to improve symptoms have not been validated as an effective treatment because of limited qualitative evidence from clinical trials and, thus, remains an uncertain therapy option [14,15]. Therefore, further studies are warranted in broad and specific populations, including different geographical regions, and over extended periods particularly because the current accepted therapeutic strategies in IBS are inadequate [16].

The effects of probiotics need to be verified as beneficial and with the accepted safety profile for healthy individuals [17]. Some recent studies have reported the biological and therapeutic efficacy of probiotics on healthy individuals and patients [6,18]. However, these effects are strain-specific and are sometimes controversial, necessitating further investigation to affirm their health benefits [1]. Moreover, recently it was shown that acid and bile resistance are incredibly variable depending on the composition of the formulations (the contained species and strains) [19]. Except for the formulations and compositions, the specific features of the patients (intestinal peristalsis, gut transit time, etc., can influence how much of the probiotics reach the target areas (i.e., colon). 

In our study, we aimed to investigate the effects of a 21-day administration of a product containing *Lactobacillus* strains, larch arabinogalactan, and colostrum on the primary pro- and anti-inflammatory cytokines in the intestinal mucosa and natural killer (NK) cells in the peripheral circulation in both healthy individuals and patients with IBS. In addition, we intended to assess IL-6, TNF-α, IFN-γ, IL-10, IL-17A, and secretory IgA in fecal samples because of the constant interactions between the microbiome and mucosal surfaces, where the outcomes of these interactions could be evaluated [1], and NK cells in the peripheral circulation as evidence for the indirect effects of some potentially probiotic strains on these cells in the circulation. Furthermore, we assessed differences between the study groups at baseline regarding some immunological parameters in the mucosa and peripheral blood (i.e., cytokine levels and NK cells, respectively).

## 2. Materials and Methods

### 2.1. Study Design

In this single-center, blinded trial, we enrolled 40 individuals (20 patients with IBS and 20 healthy individuals). We collected participants’ blood and fecal samples before and after a 21-day administration (2 × 2 capsules daily) of the formula. We tested the blood samples of subjects for total and activated NK cells using the flow cytometric analysis and fecal samples for cytokines IL-6, TNF-α, IFN-γ, IL-10, IL-17A, and fecal IgA using ELISA.

All participants were asked to maintain a daily diary of their sensations or symptoms for the three-week study period. Our study protocol clearly specified that both healthy individuals and patients should not take any medicine or additives, such as antibiotics and commercially available probiotics, as well as agents that could affect the gut motor or absorptive capacity, including laxatives and anti-diarrheal agents, which could alter the intestinal function or flora, for at least one month before and during the trial. Dairy products and probiotic-containing foods, such as yogurt, were also prohibited one month before and during the study. Patients and health people were followed-up regularly by phone and on their visits to the clinic.

All subjects gave their informed consent for inclusion before they participated in the study. The study was conducted in accordance with the Declaration of Helsinki, and the protocol was approved by the Ethics Committee of Medical University of Sofia, Bulgaria (Protocol No 1034/17.11.2015).

### 2.2. Subjects of the Study

In this study, we enrolled 20 patients with IBS (6 males and 14 females; mean age, 37 ± 2 (range, 24–52) years) and 20 healthy individuals (7 males and 13 females; mean age, 35 ± 2 (range, 19–66) years), all of whom were Caucasian. We diagnosed patients with IBS as per the Rome III criteria after conducting necessary studies to exclude physical, radiological or endoscopic abnormalities or laboratory findings indicating an organic disease. All of our IBS patients mostly had diarrhoea (IBS-D). The exclusion criteria included any clinically significant systemic disorder, inflammatory bowel disease, immunodeficiency, prior abdominal operation, or pregnancy. We recruited patients from the Gastroenterology Clinic at the University Hospital ‘Tsaritsa Joanna-ISUL’ (Sofia, Bulgaria).

The healthy individuals were provided by Lactina Ltd. as follows: Eight employees of the company and 12 paid volunteers who were negative for autoimmune disease markers (such as anti-nuclear antibodies, rheumatoid factor, and anti-neutrophil cytoplasmic antibodies), severe organic diseases (excluded by, but not limited to, the following examinations: Complete blood count, ESR, standard biochemical studies, CRP, microbiological fecal study, fecal calprotectin, and abdominal ultrasound) and psychiatric diseases.

Among the patients, two were smokers, and among the healthy controls, four were smokers. No participants took antibiotics, probiotics, laxatives, or anti-diarrhoeal agents at least one month before and during the study.

No one of the study subjects dropped out of the study.

### 2.3. Lactobacillus Blend

We used a *Lactobacillus* blend, including five *Lactobacillus* strains previously selected as per their antimicrobial activity and antibiotic susceptibility [20]. One capsule of the blend (400 mg) comprised (i) 67.5% freeze-dried *Lactobacillus* strains (*Lactobacillus acidophilus* LLA-10, *L. helveticus* LLH-108, *L. rhamnosus* LLR-L1, *L. fermentum* LLF-01, and *L. bulgaricus* LLB-06 in the ratio of 1:1:1:1:0.5; Lactina Ltd.), (ii) 37.5% arabinogalactan from Larix dahurica (Ametis JSC, Blagoveshchensk, Russia) and (iii) 20% skim bovine colostrum, min. 20% IgG APS60-20I (ApsBiogroup, Phoenix, AZ, USA). Each 400 mg capsule contains 270 mg active substances in accordance with the ratio 1:1:1:1:0.5 as follows: *L. rhamnosus* LLR-L1–100 million, *L. acidophilus* LLA-10–50 million; *L. helveticus* LLH-108–50 million, *L. fermentum* LLF-01–50 million, and *L. bulgaricus* LLB-06–5 million, as total of active substances 250 million. 

According to the manufacturer, skim bovine colostrum APS 60-20I contained 60% protein minimum and 20% IgG minimum. It was an agglomerated, instantized, water-soluble, pasteurized, reduced fat, colostrum powder produced from colostrum collected only from the first milking colostrum. APS 60-20I was processed both at low pressures and temperatures and was spray-dried using indirect steam to maintain maximum bio-activity.

Each capsule of the blend contained 250 million viable cells at the time of manufacture. We partitioned the blend in gastro-resistant capsules (DRcapsTM capsules, Capsugel, Morristown, NJ, USA) for oral administration.

### 2.4. Specimen Collection and Preparation

All participants collected their fecal material using a spatula in a provided tube. Every sample was immediately prepared after transportation to the laboratory. We prepared the fecal samples as follows: 1 g fecal matter was homogenized in 1000 mL of phosphate-buffered saline, followed by centrifugation at 20,000× *g* for 15 min at 4 °C. The supernatants of the fecal extracts were then frozen at −80 °C. Before testing, all samples were entirely thawed at room temperature.

Furthermore, after obtaining informed consent, we collected each participant’s whole peripheral blood (EDTA, 1–3 mL), which was then immediately transported to the laboratory for flow cytometry.

### 2.5. Flow Cytometry

We tested the blood samples for the total number and percentage of activated NK cells using flow cytometry (BD FACSCalibur; BD, USA) with monoclonal antibodies (anti-CD3 FITC, anti-CD16/56 PE, and anti-HLA-DR PE Cy5; BD, USA). We defined NK cells as CD3−CD16/56 + HLA-DR+/−. Flow cytometry was performed in the Laboratory of Clinical Immunology, St. Ivan Rilski University Hospital (Sofia, Bulgaria).

### 2.6. Enzyme Immunoassays

We performed ELISA in the fecal samples to detect cytokines IL-6, TNF-α, IFN-γ, IL-10 and IL-17A (each with a Human ELISA kit; Gen-probe Diaclone SAS, Besançon, France) and fecal IgA (Secretory IgA ELISA; Immuchrom GMBH, Heppenheim (Bergstraße), Hessen, Germany). ELISA was performed at the Laboratory of Clinical Immunology, St. Ivan Rilski University Hospital.

### 2.7. Statistical Analysis

We statistically analyzed the row data using parametric and non-parametric tests (Software Package for Statistical Analysis SPSS^®^, v.19 (IBM). Moreover, differences were considered statistically significant at *p* < 0.05.

## 3. Results 

### 3.1. NK Cells in the Peripheral Blood

We observed no difference in the total number of NK cells in the peripheral blood between groups either at baseline or after supplementation (*p* > 0.05; Table 1). However, the percentage of HLA-DR+ NK cells was significantly higher in patients with IBS than in healthy individuals both at baseline and after supplementation (*p* < 0.001; Table 1). While we did not observe a significant difference between the total number of NK cells before and after supplementation in patients with IBS, the rate of activated NK cells was higher after supplementation than before in both healthy individuals (*p* = 0.03) and patients with IBS (*p* = 0.681) (Table 1).

### 3.2. Cytokines and IgA in Fecal Samples

In patients with IBS, we observed a decline in IL-6, IFN-γ, TNF-α, and secretory IgA levels in fecal samples and, simultaneously, an increase in IL-10 and IL-17A levels after supplementation. However, these findings were not statistically significant (Table 2).

In our previous studies in healthy controls, we observed a significant decline in the levels of two of the assessed cytokines, IL-6 and IFN-γ, in fecal samples after supplementation (*p* < 0.001) [21]. The data of healthy subjects is compared with IBS patients (Figure 1). However, two of the cytokines, TNF-α and IL-10, along with fecal IgA, also presented with lower levels than those at baseline, although not statistically significant. In addition, IL-17A levels did not dramatically alter in healthy individuals but slightly increased after supplementation in the IBS group (*p* > 0.05) (Figure 1).

Comparison of cytokines levels in fecal samples at baseline revealed that IL-6 (*p* < 0.001), IFN-γ (12 times higher) (*p* < 0.001), TNF-α (*p* > 0.05), and IL-10 (22 times higher) (*p* > 0.05) levels considerably varied between healthy individuals and patients with IBS. The differences remained for the cytokines mentioned above, except for IL-6, after supplementation (Table 2).

### 3.3. Clinical Assessment of Patients with IBS

In this study, no participant reported an adverse effect during the administration of the product. We documented a clinical improvement of symptoms in 65–75% of patients with IBS and the complete resolution of initial symptoms in five of the 20 patients. Besides, we observed a marked reduction in the three main symptoms of patients with IBS, namely diarrhea, abdominal pain, and abdominal discomfort (Table 3), with the latter comprising bloating, distension, a sensation of incomplete evacuation, the passage of gas, straining and bowel habit satisfaction.

The calprotectin levels also decreased from 161 µg/mL to 77.1 µg/mL (*p* = 0.004). Moreover, in 13/20 patients, the levels dropped below the upper reference range.

## 4. Discussion

Probiotics exert beneficial effects on the intestinal mucosa via various mechanisms, including the inhibition of the growth and binding of pathogenic bacteria, amelioration of the epithelial barrier function, and modulation of the local and systemic immunity of hosts [11]. Through these alterations, specific probiotic strains have been demonstrated to enhance innate immunity, phagocytosis, and NK cell activity primarily; however, acquired immunity remains unaffected [22]. Among the available strains, *Lactobacillus* strains have been the most investigated, which have exhibited beneficial effects on gut health [23]. Thus, we used a blend of five *Lactobacillus* strains, namely *L. acidophilus, L. helveticus, L. rhamnosus, L. fermentum*, and *L. bulgaricus*. Reportedly, *L. acidophilus* and *Bifidobacterium breve* increase the activity of NK cells and enhance their activation [24]; however, a more extensive variety of *Lactobacillus* strains has been known to increase the number and activity of NK cells because of an interaction between NK and dendritic cells [22]. Furthermore, some studies have reported that the activity of NK increased overall and was not specific to any individual strain of commercial lactobacilli or bifidobacteria [25].

In this study, the number of NK cells in the peripheral blood did not differ after supplementation in healthy individuals and patients with IBS, despite the significantly (healthy individuals) and slightly (patients with IBS) increased percentage of the activated fraction of NK cells (CD3−CD16/56 + HLA-DR+). We anticipated that by increasing the percentage of activated NK cells, the treatment positively enhanced the antiviral activity of the immune system. Nevertheless, the percentage of activated NK cells was three times higher in patients with IBS than in healthy individuals at baseline, suggesting the initial innate immune activation in IBS as a feature of this condition.

Various cytokines (i.e., INF-γ) increase the number of NK cells and stimulate their activation. In contrast, NK cells cytotoxicity and cytokine production are intimately associated with perforin, granzymes, and the TNF-α family of ligands [22]. The effect of probiotics on NK cells may be attributed to secreted specific cytokines that regulate the activity of NK cells, leading to a form of cross-talk between innate and adaptive immunity [22]. Regarding cytokine production, the specific strains could evidently significantly increase IL-1β, IL-6, IL-10, TNF-α levels, along with GM-CSF, MIP-1α [25]. Previous studies showed a significant decline in the levels of two of the assessed cytokines, IL-6 and IFN-γ, in fecal samples after supplementation; however, we obtained this result only in healthy individuals [21]. Both cytokines, primarily accepted as “pro-inflammatory,” have been shown to possess multiple and controversial properties. While IL-6 might enhance immunoglobulin synthesis, including in the gut mucosa, IFN-γ induces Th1 responses and cytotoxicity by affecting enterocytes to present antigens in the context of HLA-II molecules [26,27]. 

Our results show that supplementation has a systemic effect by affecting IL-6, IFNγ, and NK cells expressing HLA-DR. A possible explanation of why this effect is realized only in healthy people might be the different systemic cytokine background in healthy people and patients with IBS [28], based on which the supplementation works. It is logical to assume that the supplementation affects IL-6, IFNγ, and DR expression on NK cells differently on the base of distinct cytokine backgrounds.

In conclusion, the assessed product containing lactobacilli favors local immunosuppression by decreasing IL-6 and IFN-γ levels and restoring immune tolerance in the gut mucosa, which is a desired state of the gastrointestinal tract when frequently exposed to a large number of microbial and food antigens. However, we also documented a non-significant decrease in TNF-α and IL-10 levels in healthy individuals and IFN-γ and TNF-α levels in patients with IBS. Remarkably, in patients with IBS, IL-10, and IL-17A levels were elevated after supplementation (Table 2). 

Unlike TNF-α, which is an example of a pro-inflammatory cytokine, IL-10 is challenging to be definitely determined. It is one of the cytokines with the most pleiotropic action. Classically described as anti-inflammatory because it leads to the formation of tolerogenic dendritic cells and Tregs. However, IL-10 is a typical Th2 cytokine that stimulates the humoral immune response. In addition, IL-10 is also involved in the processes of apoptosis [29]. 

For this reason, the effect of its reduction in healthy people, although non-significant in our study, could hardly be treated unambiguously. In all cases, it can be speculated that supplementation with probiotics also affects this pleiotropic and key for the functioning of the immune system cytokine. Due to the pleiotropic nature of IL-10, the biological meaning of this decline should not be interpreted per se, but in the context of other clinical and immunological changes.

Considering the relatively small size of our study cohort, the results might have failed to demonstrate differences between groups or considerable alterations of cytokine levels at baseline and after supplementation.

Besides, speculation regarding the pathogenesis of IBS and its symptoms might have resulted from the presence of low-grade inflammation. Such protuberance between the increased production of pro-inflammatory cytokines and decreased production of anti-inflammatory cytokines has been previously demonstrated [30]. Thus, reversing the imbalance between the pro- and anti-inflammatory cytokines besides declining the bacterial overgrowth in the small bowel by increasing the mass of beneficial bacteria in the digestive tract is the indisputable beneficial effect of probiotics in IBS [12]. Some studies have reported that patients with IBS who underwent the successful eradication of bacterial overgrowth exhibited improvement in their symptoms [11]. A leading “gas-related” symptom reported in patients with IBS could be alleviated by a reduction in bacterial fermentation and modulation of the composition (biodiversity and number) of the flora [31]. In this study, we observed a considerable improvement of clinical symptoms, such as diarrhea (75% of patients), abdominal pain (65%), and abdominal discomfort (65%) including bloating/distension (Table 3), as we have also shown previously [32]. A study reported that TNF-α, IL-1, and IL-6 levels in the peripheral blood mononuclear cells correlated with the symptom severity in IBS, including the intensity and frequency of painful events and motility-related symptoms [33]. However, in our study, we did not establish a correlation between the cytokine levels in fecal samples and clinical signs.

Some studies have demonstrated the overall improvement of IBS symptoms with probiotics, rather than a specific improvement in the bowel or other functions [16]. In contrast, other studies have reported that probiotics are more active on single symptoms than on the entire IBS manifestations [12]. However, it is difficult to compare the available studies because of variations in the study design, probiotic strains used, doses administered, and formulation. Specific strains, such as Bifidobacterium, have been proven to exert a beneficial effect in the relief of IBS symptoms, either as a single agent or in combination with other probiotics. However, only multi-strain trials have established a clinically meaningful improvement in the quality of life of patients with IBS. A possible elucidation is associated with the multi-strain challenge, which causes a higher reduction of the inflammation-modulated genes [10]; this served as the background for our team to select a probiotic blend with five *Lactobacillus* strains rather than investigating the immunomodulating properties of a single strain.

Another non-digestible part of the investigated product—larch arabinogalactan—specifically alters the composition and activity of beneficial microbiota, including bifidobacteria and lactobacilli [1,34,35]. The fermentation of larch arabinogalactan by the resident colonic microflora results in the production of SCFA, butyrate, acetate, and propionate, the latter two being predominantly produced by arabinogalactan fermentation [36]. In addition to their essential role as fuels for intestinal epithelial cells, these fatty acids form a vital link in the cross-talk between the gut microbiota and the immune system [37]. Reportedly, SCFAs play roles in regulating the leukocyte production of cytokines (such as TNF-α, IL-2, IL-6, and IL-10) and eicosanoids as well as in inducing the differentiation and expansion of regulatory T cells in the colon and peripheral circulation [38,39]. Moreover, different studies have suggested that larch arabinogalactan increases the activity and cytotoxicity of NK cells primarily by an augmented release of IFN-γ [40,41].

As mentioned earlier, we observed an increase in the number of activated NK cells in the peripheral blood after supplementation, but we did not assess their cytotoxicity. Besides, we observed altered IFN-γ levels (decreased after supplementation) in the fecal samples. Still, we did not evaluate these in the blood samples of our patients, which could illustrate the lowering of cytokine levels in the mucosa. Apparently, the steady-state of the mucosa is the suppression of the local inflammatory response by decreasing the level of pro-inflammatory cytokines such as IFN-γ or TNF-α. Furthermore, skimmed colostrum increases the cytotoxicity of NK cells but also supplies the mucosal immune system with growth and nutritious factors, immunoglobulins, lactoferrin, cytokines, and proline-rich polypeptides with antimicrobial and anti-inflammatory properties [42]. In our study, the last investigated parameter in the fecal samples of participants was secretory IgA levels. We observed some changes in the IgA levels towards lowering, but these alterations were not significant. However, some studies have reported a beneficial effect of probiotics on secretory IgA levels in the gut mucosa towards increase, particularly in young adults [22].

This study has some limitations. The small size of the study population and the relatively short-term treatment might have failed to detect substantial alterations of the number of NK cells in the peripheral circulation, cytokine and IgA levels in the fecal samples and effects of potentially probiotic strains included in the formula on some symptoms of patients with IBS. Moreover, we obtained results that at baseline the IL-6, IFN-r and TNF-a levels in patients were significantly lower than that in healthy controls. However, it needs to be stressed that the duration of this study corroborated that of other recent studies, and further follow-up of the study cohort is necessary to assess any long-term effect of this treatment. We did not take into account the smoking status of the participants. However, due to the small number of smokers in our study, we believe that this had not a significant impact on the results. Additionally, in our study, we did not analyze the microbiota profile in the fecal samples. However, this could be an opportunity to conduct future research.

## 5. Conclusions

This study established the immunomodulating effect of the oral administration of a formula containing *Lactobacillus* strains, arabinogalactan, and colostrum towards increasing the number of activated NK cells in the peripheral blood as well as decreasing IL-6 and IFN-γ levels in the fecal samples, both in healthy individuals and patients with IBS. These results are speculative for a possible prophylactic effect of the used formula on healthy individuals attained by the inducing system antiviral impact along with a trend for the local immune tolerance in the gut where it is the most desirable state. Regarding patients with IBS, well-designed, extensive studies are warranted to resolve the issues such as the most effective dose of the formula components and the duration of treatment, whether patients should be treated only for specific IBS symptoms or for the overall improvement of the quality of life, and cost-effectiveness analysis and safety profiles.

## Figures and Tables

**Figure 1 nutrients-12-01706-f001:**
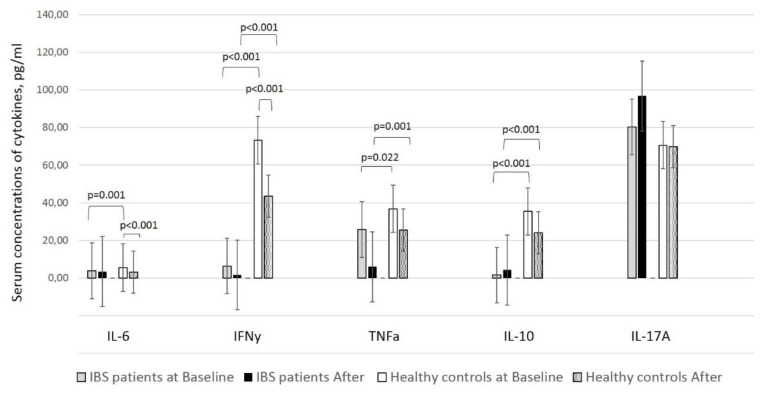
Cytokines and fecal IgA levels in fecal samples of healthy controls and patients with IBS before and after the supplementation of a product containing lactobacilli supplemented with larch arabinogalactan and colostrum. Results are presented as concentration (pg/mL or µg/mL) ± SD (range).

**Table 1 nutrients-12-01706-t001:** Total number and percentage of activated natural killer (NK) cells at baseline and after oral administration of a product comprising lactobacilli supplemented with larch arabinogalactan and colostrum in healthy controls and patients with IBS.

	Patients with IBS			Healthy Controls			IBS vs. Healthy	
	Baseline	After	*p*, Baseline vs. After	Baseline	After	*p*, Baseline vs. After	*p*, Baseline	*p*, After
**Total number of NK cells**	15.3 ± 1.5 (5.5–31.1)	14.2 ± 1.5 (6–28.7)	0.269	16.1 ± 1.2 (8.1–25.5)	14.6 ± 1.2 (5.1–25.7)	0.073	0.673	0.845
**HLA-DR^+^ NK cells (%)**	14.5 ± 0.9 (9.3–23.6)	15 ± 1.1 (7.6–26.4)	0.681	4.9 ± 0.6 (1.3–9.9)	7.1 ± 0.7 (2.6–13.5)	0.03	<0.001	<0.001

Results are presented as mean ± SD (range).

**Table 2 nutrients-12-01706-t002:** Cytokines and IgA levels in fecal samples of healthy controls and patients with irritable bowel syndrome (IBS) at baseline and after supplementation of a product comprising lactobacilli supplemented with larch arabinogalactan and colostrum.

	Patients with IBS			Healthy Controls			IBS vs. Healthy	
	Baseline	After	*p*, Baseline vs. After	Baseline	After	*p*, Baseline vs. After	*p*, Baseline	*p*, After
**IL-6, pg/mL**	3.8 ± 0.4 (0.23–8.43)	3.5 ± 0.3 (1.1–6.7)	0.587	5.6 ± 0.2 (3.9–6.75)	3.1 ± 0.1 (2.3–5.5)	<0.001	0.001	0.319
**IFN-γ, pg/mL**	6.4 ± 3.9 (0–69.5)	1.7 ± 0.9 (0–15.8)	0.499	73.2 ± 2.6 (55.5–92.5)	43.6 ± 3.9 (28.3–99)	<0.001	<0.001	<0.001
**TNF-α, pg/mL**	25.8 ± 14 (0–214)	6.0 ± 4.4 (0–82.5)	0.173	36.8 ± 9.6 (0–116.5)	25.6 ± 7.5 (0–103.3)	0.215	0.022	0.001
**IL-10, pg/mL**	1.6 ± 0.7 (0–11.3)	4.3 ± 2.7 (0–53)	0.441	35.4 ± 8.4 (2.5–178)	24 ± 5 (7.8–108.8)	0.135	<0.001	<0.001
**IL-17A, pg/mL**	80.2 ± 26.6 (0–478)	96.7 ± 36.3 (0.3–613.1)	0.313	70.6 ± 25.5 (0–387)	69.9 ± 23 (0–100.1)	0.795	0.447	0.465
**Secretory IgA, μg/mL**	3671.2 ± 738 (0–8406.9)	3334.4 ± 721 (0–8458.75)	0.691	3001.5 ± 599 (207.5–8355.63)	2806.4 ± 622.3 (183.75–8818)	0.747	0.485	0.583

Results are presented as mean concentration (pg/mL or µg/mL) ± SD (range).

**Table 3 nutrients-12-01706-t003:** Clinical improvement of patients with IBS after supplementation.

Symptoms	At Baseline		After Supplementation		Significance, *p*
	Number	%	Number	%	
Diarrhoea	20	100	5	25	<0.05
Abdominal pain	19	95	7	35	<0.05
Abdominal discomfort	17	85	5	35	<0.05

Results are presented as a number/% of patients with a declared presence of the described symptoms.

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
