# Peer review of "Lactobacilli Supplemented with Larch Arabinogalactan and Colostrum Stimulates an Immune Response towards Peripheral NK Activation and Gut Tolerance"

_nutrients, 2020, doi:10.3390/nu12061706_

Round 1

Reviewer 1 Report

 Velikova  assessed the use of Lactobacilli supplemented with larch arabinogalactan 2 and colostrum on a variety of different cytokines pre and post a trial of the cocktail and also found some symptomatic improvements in IBS patients after supplementation. I have the following i think should be addressed to help improve the manuscript.

Major

  1. Please account for other important variables
    1. smoking status
    2. ethnicity
    3. other co-morbidities
    4. other drug use
  2. Please sub-categorise your IBS patients- how many were diarrhoea predominant/ constipation predominant.
  3.  Limitations- lack of confounding variables accounted for and lack of IBS sub categorisation

minor

  1. "However, the growing prevalence of the Western diet as a part of the modern lifestyle 40 may favor a transition in the microbial composition, leading to dysbiosis" - Please discuss other pressures that may influence the gut microbiome other than diets to include the environmental factors, smoking, lifestyle, drug use etc

2. Abinogalactans are a source of dietary fibers and therefore i would discuss what are pre-biotics and some data around their benefits

3. have garnered ( suggest changing to have gathered)

4. The effects of probiotics need to be verified as beneficial and with the accepted safety profile 75 for healthy individuals [15]. Some recent studies have reported the biological and therapeutic 76 efficacy of probiotics on healthy individuals and patients [5,16]. However, these effects are strain-77 specific and are sometimes controversial, necessitating further investigation to affirm their health 78 benefits [1]. -

I would add a comment about how much probiotic actually gets into the target areas (colon) 

5. Besides, we tested the blood samples for total 94 and activated NK cells using the flow cytometric analysis and fecal samples for cytokines IL-6, 95 TNF-α, IFN-γ, IL-10, IL-17A, and fecal IgA using ELISA. 

Please rephrase - besides is a strange way to begin a sentdnce

6. Please provide some more details on colostrum use- where did you get this? how was it made? How was it administered

7. were there any drop outs if so how were these analysed

8. How did you assess compliance of taking the probiotics?

9. Please consider more speculation on why IL-10 went down in healthy controls

Author Response

Dear Editor,

Dear Reviewers,

Thank you for your time to revise our Manuscript ID: Nutrients-807546, entitled "Lactobacilli supplemented with larch arabinogalactan and colostrum stimulates an immune response towards peripheral NK activation and gut tolerance." We acknowledge that our paper might have some issues in conformity with the following comments.

Reviewer 1

Open Review

( ) I would not like to sign my review report

(x) I would like to sign my review report

English language and style

( ) Extensive editing of English language and style required

(x) Moderate English changes required

( ) English language and style are fine/minor spell check required

( ) I don't feel qualified to judge about the English language and style

  • Thank you for your valuable comments and the overall evaluation of the manuscript. The paper was proofread and edited by a professional company of English editing, and we have a certificate that can be uploaded with the revised manuscript. However, we have revised the text once again to eliminate some remaining mistakes.

Comments and Suggestions for Authors

Velikova assessed the use of Lactobacilli supplemented with larch arabinogalactan and colostrum on a variety of different cytokines pre and post a trial of the cocktail and also found some symptomatic improvements in IBS patients after supplementation. I have the following I think should be addressed to help improve the manuscript.

  • Thank you for the evaluation and for the valuable suggestions to improve the paper.

Major

Please account for other important variables

smoking status

ethnicity

other co-morbidities

other drug use

  • Thank you for the valuable comment. We agree that these factors can influence the results significantly. Therefore, we provided the missing information in the Material and methods section for both healthy persons and patients – the Caucasian race (lines 128) and the smoking status (lines 140). However, the most potent factors, such as other comorbidities and drug use, were considered as excluding factors, as it was stated in the Material and methods section (lines 118-120, 131-132, and 135-139).

Please sub-categorise your IBS patients- how many were diarrhoea predominant/ constipation predominant.

  • The referee is right to point out that there is a need to determine the IBS patients according to these symptoms/forms because it can impact the results. All of our IBS patients were IBD-D predominant. We have added this information accordingly (line 130).

 Limitations- lack of confounding variables accounted for and lack of IBS sub categorization

  • We agree with the referee that these are serious limitations of the study and have to be acknowledged in the limitations of the study.
  • However, since the confounding factor - smoking status, was not accounted for in the analysis for both healthy people and patients, we have added it to the limitations (lines 349-350). The other factors were accounted before the study and analysis; thus, we have added the missing information to the Material and method sections as exclusion criteria (line 140).

Minor

"However, the growing prevalence of the Western diet as a part of the modern lifestyle 40 may favor a transition in the microbial composition, leading to dysbiosis" - Please discuss other pressures that may influence the gut microbiome other than diets to include the environmental factors, smoking, lifestyle, drug use, etc

  • We agree with the referee that more factors are influencing the microbial composition in addition to the diet; therefore, we provided some information that other factors are essential as well and added a reference (lines 43-46).
  1. Arabinogalactans are a source of dietary fibers, and therefore I would discuss what are pre-biotics and some data around their benefits
  • Thank you for the valuable comment. We added the missing information along with a new reference (lines 57-64).
  1. have garnered ( suggest changing to have gathered)
  • Thank you for the valuable note. We have changes the word with the more suitable one that you have suggested (line 69).
  1. The effects of probiotics need to be verified as beneficial and with the accepted safety profile for healthy individuals [15]. Some recent studies have reported the biological and therapeutic efficacy of probiotics on healthy individuals and patients [5,16]. However, these effects are strain-specific and are sometimes controversial, necessitating further investigation to affirm their health benefits [1].

I would add a comment about how much probiotic actually gets into the target areas (colon)

  • The referee is right to point out that this is critical for probiotics because it determines the benefits for the gut and the host. We have added some comments on this critical issue and cited a new reference (lines 92-98).
  • However, given that we used gastro-resistant capsules, the probiotic substance was expected to reach at least the small intestine with unchanged activity.
  1. Besides, we tested the blood samples for total and activated NK cells using the flow cytometric analysis and fecal samples for cytokines IL-6, TNF-α, IFN-γ, IL-10, IL-17A, and fecal IgA using ELISA.

Please rephrase - besides is a strange way to begin a sentence

  • Thank you for the constructive critic. We have revised the text by removing the word "besides" as follows:

We tested the blood samples for total and activated NK cells using the flow cytometric analysis and fecal samples for cytokines IL-6, TNF-α, IFN-γ, IL-10, IL-17A, and fecal IgA using ELISA (line 110).

  1. Please provide some more details on colostrum use - where did you get this? How was it made? How was it administered
  • Thank you for the valuable note. We have added this information in the Material and methods section (lines 151-155)
  1. were there any drop outs if so how were these analysed
  • Thank you for the note. Indeed, we included in our study 20 healthy people and 20 patients with IBD with the idea to include more, if more than 5 people dropped out from the study for lack of compliance or other reasons. However, no one of the tested subjects dropped out of the study, probably due to the relatively small number of subjects that facilitated the control and compliance.
  • Therefore, the final analysis was performed on all the included subjects. We have added this statement in the Material and Method section (line 143)
  1. How did you assess compliance of taking the probiotics?
  • We support the referee's assertion that the compliance of taking the probiotics is of great importance for the design and for obtaining valid and reproducible results. Thus, we required patients and healthy people to maintain a daily diary of their sensations or symptoms for the 3-week study period, as it was stated in the Material and methods section (line 114-115). We also followed-up them regularly by phone and on their visits to the clinic.
  • We have added the information for the compliance in the Material and methods section (lines 120-121).
  1. Please consider more speculation on why IL-10 went down in healthy controls
  • There might indeed be more speculations on the reduction of IL-10 in healthy persons after supplementation, although non-significant. Therefore, we tried to add more comments and possible explanations for that result. We have added an additional reference to support our discussion on this critical issue (lines 276-285)

Reviewer 2 Report

  1. In Abstract, please delete the p value.
  2. Why only healthy individuals had a significant decline in IL-6 and IFN-γ levels and HLA-DR+ NK cells after supplementation instead of patients?
  3. In Study design, did the authors prohibit the subjects specifically to eat yogurt that could be considered as dairy resource but affects intestinal microbiota (Line 97)?
  4. Why didn’t analyze the microbiota profile in the fetal samples?
  5. In Result section, add P>0.05 in the end of the non-significantly changed results.
  6. In Table 2, why IL-6, IFN-r and TNF-a in patients was significantly lower than that in healthy control in the baseline?

Author Response

Dear Editor,

Dear Reviewers,

Thank you for your time to revise our Manuscript ID: Nutrients-807546, entitled "Lactobacilli supplemented with larch arabinogalactan and colostrum stimulates an immune response towards peripheral NK activation and gut tolerance." We acknowledge that our paper might have some issues in conformity with the following comments.

Reviewer 2

Open Review

(x) I would not like to sign my review report

( ) I would like to sign my review report

English language and style

( ) Extensive editing of English language and style required

( ) Moderate English changes required

(x) English language and style are fine/minor spell check required

( ) I don't feel qualified to judge about the English language and style

Yes      Can be improved        Must be improved       Not applicable

Does the introduction provide sufficient background and include all relevant references?

( )        (x)       ( )        ( )

Is the research design appropriate?

( )        ( )        (x)       ( )

Are the methods adequately described?

( )        (x)       ( )        ( )

Are the results clearly presented?

( )        ( )        (x)       ( )

Are the conclusions supported by the results?

( )        ( )        (x)       ( )

  • Thank you for your valuable comments and evaluation of the manuscript.

Comments and Suggestions for Authors

In Abstract, please delete the p value.

  • Not all of our results indeed reached statistical significance. However, we would be delighted if we can keep the p-values in the abstract. The reason is that we believe that this makes our manuscript more attractive and reliable for the audience.

Why only healthy individuals had a significant decline in IL-6 and IFN-γ levels and HLA-DR+ NK cells after supplementation instead of patients?

  • The referee is right to point out that these results are surprising. Nevertheless, we were also stunned, but we report the obtained results. The reasons could be attributed to the different systemic cytokine background in healthy people and patients with IBS, based on which the supplementation works; thus, it is logical that the effect of supplementation on IL-6, IFNγ, and DR expression would be distinct.
  • We have commented on this issue in the discussion section and add one more reference to support our speculations (lines 265-270).

In Study design, did the authors prohibit the subjects specifically to eat yogurt that could be considered as dairy resource but affects intestinal microbiota (Line 97)?

  • We support the referee's assertion that the consumption of yogurt and other probiotic-contacting foods can be an influencing factor. This was limited to our participants one month before and during the study. We have added this clarification in the Study design (lines 118-119).

Why didn't analyze the microbiota profile in the fetal samples?

  • The referee is right to point out that analyzing the microbiota profile in fecal samples could have been improved our paper. Unfortunately, this was not intended in our study design due to many factors. However, we cannot perform additional analysis for these subjects but we stated this information in the limitations (lines 351-352)

In Result section, add P>0.05 in the end of the non-significantly changed results.

  • Thank you for the important comment. We have added the p-value at the end of all non-significant results, as well as some of the significant, where it was not stated.

In Table 2, why IL-6, IFN-r and TNF-a in patients was significantly lower than that in healthy control in the baseline?

  • We support the referee's assertion that the baseline levels of IL-6, IFN-r, and TNF-a of patients were significantly lower than the baseline levels in healthy people. It was interesting to obtain these results, as well. However, the reasons for that are in the different cytokine background (systemic serum levels or fecal levels of these cytokines) in IBS patients vs. healthy controls, and we discussed this accordingly (lines 265-269).
  • Nevertheless, the levels dropped further after probiotic supplementations, although not significantly.

Round 2

Reviewer 2 Report

1.     No statistically significant difference should consider no difference, so whenever it says (significantly) change, it changed significantly (P<0.05).

2.     As of the significantly different baseline levels of cytokine in healthy subjects and patients, did the authors think about if it was the bias caused by the small samples or other reasons? Did the authors try to check the result differences by doing subgroup analyses? Did the authors think if the results would be repeatable?

Author Response

Dear Editor,

Dear Reviewers,

Thank you for your time to revise once again our Manuscript ID: Nutrients-807546-R1, entitled "Lactobacilli supplemented with larch arabinogalactan and colostrum stimulates an immune response towards peripheral NK activation and gut tolerance." We have revised all the issues in conformity with the following comments in Round 2 of the revision process.

Reviewer 2

  1. No statistically significant difference should consider no difference, so whenever it says (significantly) change, it changed significantly (P<0.05).
  • Yes, we agree completely with this comment and we believe that we have corrected all the issues regarding this in the previous revision.
  1. As of the significantly different baseline levels of cytokine in healthy subjects and patients, did the authors think about if it was the bias caused by the small samples or other reasons? Did the authors try to check the result differences by doing subgroup analyses? Did the authors think if the results would be repeatable?
  • Yes, we agree that the baseline cytokine levels in healthy people and patients differed significantly. However, other studies reported this observation, as well (we have cited some of them in the discussion, like the reference 28, in the previous revision).
  • Yes, we acknowledge that the sample size can affect the results, as well as other factors, which we stated this in the limitations. However, we did not perform subgroup analysis, where it would be insufficient for some statistical analyses due to even smaller numbers of the subjects.
  • Yes, we do believe that the results are repeatable, because our design and methodology were validated and approved.
